# Thromboprophylaxis in Pregnant Women with COVID-19: An Unsolved Issue

**DOI:** 10.3390/ijerph20031949

**Published:** 2023-01-20

**Authors:** Valentin Nicolae Varlas, Roxana Georgiana Borș, Mihaela Plotogea, Madalina Iordache, Claudia Mehedințu, Monica Mihaela Cîrstoiu

**Affiliations:** 1Department of Obstetrics and Gynecology, Filantropia Clinical Hospital, 011171 Bucharest, Romania; 2Faculty of Medicine, “Carol Davila” University of Medicine and Pharmacy, 37 Dionisie Lupu St., 050451 Bucharest, Romania; 3Department of Obstetrics and Gynecology, Nicolae Malaxa Clinical Hospital Bucharest, 022441 Bucharest, Romania; 4Doctoral School, “Carol Davila”, University of Medicine and Pharmacy, 4192910 Bucharest, Romania; 5Department of Obstetrics and Gynecology, University Emergency Hospital Bucharest, 050098 Bucharest, Romania

**Keywords:** COVID-19, thrombosis, antiplatelet, anticoagulant, pregnancy, low-molecular-weight heparin, thromboprophylaxis

## Abstract

SARS-CoV-2 infection in pregnant women is of growing interest due to controversy over the use of antiplatelet and/or anticoagulant drugs during pregnancy and postpartum. Pregnant women are susceptible to develop severe forms of viral infections due to pregnancy-related immune alterations, changes in lung functions, and hypercoagulability. The association of pregnancy with SARS-CoV-2 infection can cause an increased incidence of thrombotic complications, especially in the case of patients with some genetic variants that favor inflammation and thrombosis. Compared to the general population, pregnant women may be at increased risk of thrombotic complications related to COVID-19. The lack of extensive clinical trials on thromboprophylaxis and extrapolating data from non-pregnant patients lead to major discrepancies in treating pregnant women with COVID-19. Currently, a multidisciplinary team should determine the dose and duration of prophylactic anticoagulant therapy for these patients, depending on the disease severity, the course of pregnancy, and the estimated due date. This narrative review aims to evaluate the protective effect of thromboprophylaxis in pregnant women with COVID-19. It is unknown at this time whether antiplatelet or anticoagulant therapy initiated at the beginning of pregnancy for various diseases (preeclampsia, intrauterine growth restriction, thrombophilia) offers a degree of protection. The optimal scheme for thromboprophylaxis in pregnant women with COVID-19 must be carefully established through an individualized decision concerning gestational age and the severity of the infection.

## 1. Introduction

The coronavirus disease 2019 (COVID-19) pandemic continues to be one of the most challenging problems for healthcare systems worldwide. Pregnancy is a physiological prothrombotic state, and thus, there is an increased risk of developing coagulopathy and/or thromboembolic complications associated with COVID-19. During viral outbreaks, pregnancy presents a unique increased risk of respiratory infections due to changes in immune function and adaptive physiological alterations, such as increased oxygen consumption, edema of the mucous membrane of the upper respiratory tract, and the rise of the diaphragm [1]. 

During pregnancy, a series of local and systemic immunological processes allow the implantation of a semialogenous/allogeneic fetus, maintaining the antimicrobial and antiviral protection mechanisms. The succession of pro-inflammatory (implantation and placentation, induction of labor) and anti-inflammatory (angiogenesis and fetal development) events during pregnancy define a maternal immune system with good tolerance interrelated with the endocrine system. The bidirectional influences of severe acute respiratory syndrome coronavirus-2 (SARS-CoV-2) infection in pregnancy depend on the pregnant woman’s immune status, gestational age, and association with other comorbidities at the onset of the infection [2].

The variability of the inflammatory response to COVID-19 depends on several factors, including the type of SARS-CoV-2 strain. The physiological changes during pregnancy cause an increase in D-dimers and fibrinogen, and a decrease in prothrombin time (PT), activated partial thromboplastin time (APTT), and platelets. Against the modified background of coagulation, the SARS-CoV-2 infection can lead to additional changes, represented by increased D-dimers and prolonged APTT and PT [3].

The Centers for Disease Control stated an increased risk of pregnant women with COVID-19 being hospitalized and requiring admission to intensive care units because of respiratory insufficiency and low immune response [4]. Pregnant women have a four to five times higher risk of venous thromboembolism (VTE) than non-pregnant women. The prevalence of thromboembolic events is 0.5–2.0 per 1000 pregnant women, with an increased rate of morbidity and mortality [5]. In pregnant women with COVID-19, laboratory parameters of hemostasis may overlap with those of COVID-19-associated coagulopathy (CAC), disseminated intravascular coagulation (DIC), sepsis-induced coagulopathy (SIC), thrombotic microangiopathy (TMA), and HELLP syndrome [6].

Against prothrombotic status during pregnancy, international scientific societies recommend thromboprophylaxis in pregnant women and postpartum in suspected or confirmed patients with COVID-19. However, these recommendations require validation by more clinical trials [7]. We aimed to evaluate the protective effect of thromboprophylaxis in pregnant women with COVID-19.

## 2. Materials and Methods

This study was a narrative review of the English literature using two databases, PubMed and Web of Science, between March 2020 and October 2022. The keywords that were used according to the MeSH system were “COVID-19”, “thrombosis”, “antiplatelet”, “anticoagulant therapy”, “pregnancy”, “low-molecular-weight heparin”, “thromboprophylaxis”, and articles were evaluated by two reviewers from the research team. The articles included in the study were clinical studies, case series, systematic reviews, and meta-analyses, which followed the risk factors, therapeutic regimens, and maternal-fetal prognosis regarding thromboprophylaxis during pregnancy in patients with COVID-19. The excluded articles were represented by studies written in languages other than English, that did not include pregnant women, were case reports, and those in which the study design did not include statistics. In the end, 14 articles were extracted and considered eligible based on the abstracts and full texts.

## 3. Normal Pregnancy as a Hypercoagulable State

Hypercoagulability in pregnancy is determined by increased prothrombotic factors (VII, VIII, X, XII, von Willebrand factor) and fibrinogen; in addition, there is a decreased S protein and fibrinolysis. The coagulation tests can change; thus, fibrinogen and D-dimers increase and APTT shortens throughout pregnancy [8,9].

Physiological changes in pregnancy increase the risk of complications associated with SARS-CoV-2 infection, secondary to immunosuppression, respiratory dysfunction, and hypercoagulability [10] (Figure 1).

## 4. Pregnancy Outcome in Low- and High-Risk Women Exposed to COVID-19

Despite numerous articles in the literature, information on the disease evolution and complications of COVID-19 during pregnancy is incomplete and unsystematized. No data have been found about the effects of SARS-CoV-2 infection in each trimester of pregnancy, especially in the first trimester when the risk of harm to fetuses is the highest. Some studies have shown a low incidence and low rate of neonatal complications in mothers affected by SARS-CoV-2 infection [11,12].

Mullins et al. analyzed data from a group of 4005 pregnant women from the UK, the Global Pregnancy and Neonatal outcomes in COVID-19 (PAN-COVID) study, and the American Academy of Pediatrics Section on Neonatal-Perinatal Medicine (SONPM) National Perinatal COVID-19 Registry on the effects of SARS-CoV-2 infection in pregnant women and newborns. The incidence of maternal death was 0.2–0.5%, stillbirth was 0.4–0.6%, early neonatal death was 0.2–0.3%, the risk of preterm birth (<37 weeks of gestation) was 12.0–16.1%, and neonatal SARS-CoV-2 infection was 0.9–2.0% [13]. A meta-analysis by Bellos et al. on 16 observational studies, and a study of 46 pregnant women, showed that the clinical evolution of COVID-19 in pregnant women and newborns is mild, with a low mortality rate and incidence of neonatal transmission [14,15]. The evolution of the disease is more severe in the last trimester of pregnancy, especially in elderly, obese, or low socioeconomic status pregnant women, who have an increased rate of maternal mortality, preterm birth, and hospitalization in intensive care units [12].

The vulnerability of pregnant women is also increased because of changes in the transmission rate, with the appearance of mutations and viral variants of SARS-CoV-2. An increase in the transmission rate of variants in the UK, Brazil, and South Africa was observed due to the easy binding of the virus to the human receptor angiotensin-converting enzyme 2 (ACE-2) [16,17]. Its binding achieves the mode of action of SARS-CoV-2 to target cells by ACE-2 [18]. The susceptibility of pregnant women to develop severe forms of viral infections is due to immune changes, and the increased risk of hypoxemia due to pulmonary functional changes encountered in pregnancy. The risk of acute respiratory distress syndrome is secondary to the physiological increase in plasma volume and, respectively, capillary leakage [19,20].

## 5. Pathophysiological Considerations of Endothelial Alterations, Proinflammatory Thrombotic Effects, and Immunothrombosis in COVID-19 Patients

The main sites of SARS-CoV-2 infection are pneumocytes, vascular endothelial cells, and immune cells. SARS-CoV-2 can cause endothelial damage and vasculitis secondarily to a pro-inflammatory status. The endothelial injury occurs due to direct SARS-CoV-2 infection and the action of the immune system and thrombo-inflammatory responses [21,22]. Subsequent micro- and macrovascular thrombosis occurs in the entire circulation, involving arterial, capillary, and venule beds, and determines a prothrombotic status. Thromboembolic complications in COVID-19 have a traditional VTE component and an in situ immunothrombotic component [3].

The main mechanism of action of SARS-CoV-2 is the binding of the virus to the ACE-2 receptor. In addition to the expression of ACE-2 in the epithelium of the respiratory tract, a crucial role is played by the coupling of ACE-2 expressed in vascular endothelial cells [23]. The pathophysiological triad is represented by stimulation of the renin-angiotensin system, decreased fibrinolytic activity, and increased platelet aggregation [18]. This is followed by an increase in the production of proinflammatory cytokines (interleukin-6) with an increase in vascular permeability, activation of coagulation pathways, stimulation of megakaryopoiesis, and stimulation of coagulation factors synthesis [24,25]. With the penetration of the virus into endothelial cells, both direct lesions and endothelial alterations appear, with proinflammatory thrombotic effects, by inhibiting fibrinolysis and stimulating thrombin production [22,26]. Neutrophil activation and platelet-neutrophil interrelationship lead to the appearance of immunothrombosis [3,27]. One hypothesis regarding antiplatelet therapy’s mechanism is the blockage of the positive feedback loop by decreasing the activation of neutrophils and coagulation factors, leading to a decrease in the immunothrombosis cascade [3,28].

SARS-CoV-2 infection is accompanied by a significant inflammatory reaction, which underlies the cytokine storm, evidenced by the abnormal increase in D-dimers. The prothrombin time activity (PT-act) <75%, together with elevated D-dimers and fibrin degradation products, are predictive factors for mortality rate in patients with COVID-19 [7]. The inflammatory reaction is directly proportional to the severity of the disease [7], the main mechanism being the secondary coagulopathy, initially with micro thrombosis and then with extensive thrombosis in various organs. Genetic variations modulating this process can raise the plasma levels of pro-inflammatory cytokines or inflammation markers implicated even more. Variations in the genes implicated in inflammation processes: CRP, TNFα, IL4, IL6, IL1β, and SELP increase susceptibility to the higher inflammatory response [29]. The consequence of decreased capillary blood flow is tissue hypoperfusion, with devastating consequences in multiple organs [8].

If the effect of a genetic variation hinders the synthesis pathway of anticoagulant compounds, thrombosis becomes more probable. Homocysteine metabolism plays a role in developing thrombosis due to various mechanisms (increase the expression of adhesion molecules and the blood coagulation factor V, inhibit fibrinolysis, and disrupt nitric oxide metabolism) [29].

Studies have shown that in patients with COVID-19, thrombocytopenia is not common. Moderate to severe forms of thrombocytopenia, secondary to the cytokine storm, are due to inhibition of thrombopoiesis, antithrombocyte autoimmune response, and consumption coagulopathy secondary to viral proinflammatory effects [30,31]. Other studies show thrombocytosis in moderate to severe cases, either by the pro-inflammatory action of cytokines or by the interaction between megakaryocytes and coagulation factors (von Willebrand factor) [32]. In patients with COVID-19, elevated levels of D-dimers in the lungs are thought to coexist with low fibrinolytic levels [33].

Acute viral infection with SARS-CoV-2 through inflammatory and secondary thrombotic phenomena can initiate severe antiphospholipid syndrome through endotheliosis, diffuse microvascular thrombosis due to excessive activation of the complement pathway, platelet destruction, vasospasm followed by tissue hypoperfusion [8]. Organ dysfunctions secondary to SARS-CoV-2 infection occur both in the mother and the fetus by highlighting the expression of ACE2 at the level of endothelial cells and in various fetal organs, except the kidney. This makes it impossible to detect viral RNA in the amniotic fluid due to the absence of urinary excretion of SARS-CoV-2 [34].

The reaction of the maternal body to the infection determines a placental inflammatory response characterized by increased levels of cytokines, followed by the induction of the fetal inflammatory response syndrome, with consequences on the neurodevelopment of the fetus [35]. The critical evolution of the condition of pregnant women leading to death is due to the extensive phenomena of microvascular thrombosis at the pulmonary level, immunothrombosis, and immobilization secondary to venous thromboembolism [18].

## 6. Risk of Thrombosis in Low- and High-Risk Pregnant Women Exposed to COVID-19

Pregnancy does not raise the susceptibility to COVID-19 but may worsen the clinical evolution of the disease compared to non-pregnant women of similar age. One of the most feared complications of COVID-19 is VTE. Pregnant women with COVID-19 may have an even higher risk of VTE. Pregnant women develop severe respiratory forms more often in the third trimester of pregnancy and puerperium or in cases with associated comorbidities (obesity, diabetes, asthma) [36]. In principle, the incidence of thrombosis in hospitalized pregnant patients is 2% [37,38], which is much lower than that of patients with COVID-19 who are not pregnant (21%) [1].

The severity of this condition during pregnancy and postpartum requires special recommendations regarding treating acute thrombotic events and thromboprophylaxis for those with high-risk factors. Some recommendations for treating CAC in pregnant women come from reports published in the non-pregnant population, so their use requires several specific precautions [39]. In severe cases of COVID-19, coagulopathy appears to be related to venous and arterial thromboembolic disease. Due to the polymorphism of the symptoms, COVID-19 patients develop a distinct intravascular coagulation syndrome (microvascular thrombosis and endothelitis) that requires particular diagnostic criteria [40].

The individual risk of VTE increases if additional risk factors are present. Prophylaxis should preferably be performed with low molecular weight heparin (LMWH) at the dose of high-risk prophylaxis for 6 to 14 days. These recommendations were made primarily for hospitalized patients, but VTE prophylaxis in outpatients should be based on the same criteria as in-hospital prophylaxis [15].

The PROphylaxis of ThromboEmbolism in Critical Care Trial (PROTECT) randomized 3764 patients, with an overall VTE rate of 9.1%, a deep venous thrombosis (DVT) rate of 5.5%, and a pulmonary embolism (PE) rate of 1.8%. This study showed that the administration of dalteparin or other LMWH is not superior to unfractionated heparin for preventing proximal leg DVT. The thrombosis rate was 6.6% for 327 non-critical patients [41]. Another study, based on the severity of the disease, found an increased thrombotic risk in pregnant women with severe COVID-19 than in those with milder forms (6.6% vs. 3.7%), with similar rates of DVT or PE [42]. So far, no consensus has been reached on setting the optimal dose to prevent thrombotic events. Studies argue for the standard prophylactic dose [43], or for more intense anticoagulation [44].

Several meta-analyses revealed more data that are focused on thrombotic complications of COVID-19 [45,46]. The meta-analysis performed by Noop et al. on 66 studies and over 28,000 patients reports an overall in-hospital VTE incidence of 14.1% [46]. Notably, this incidence could not be adjusted for the competing risk of mortality, nor was it indicated at which point during the disease the VTE diagnosis was confirmed [47].

McGonagle et al. proposed the term “diffuse intravascular pulmonary coagulopathy,” which occurs early by increasing D-dimers after DIC [48]. Goshua et al. have histologically confirmed the presence of extensive thrombosis in micro vascularization [21]. A possible protective effect due to prolonged anticoagulant therapy has been associated with decreased mortality [49].

Dashraath et al. showed that 2% of pregnant women with COVID-19 require assisted mechanical ventilation [50]. The International Society of Thrombosis and Hemostasis (ISTH) guideline recommends hospitalization of patients with significantly increased D-dimers, with prolonged PT, with thrombocytopenia, or with fibrinogen <2 g/L [30].

Currently, there is an important variability regarding the use of guidelines (RCOG, ACOG, CNGOF, COVID Collaborative Group Barcelona, NIH, ASH, ESVS, ISTH) (Table 1). The dose of LMWH was dependent on the severity of the disease. Thus, the standard prophylactic dose for a mild disease is over 7 days, and the therapeutic dose is based on weight for intermediate forms over 6 weeks. The anticoagulation approach was individualized for severe forms over 6-12 weeks after birth [51,52,53,54,55,56,57,58].

### 6.1. COVID-19 Associated Coagulopathy

The assessment of prothrombotic mechanisms regarding the severity of COVID-19 in pregnant women has not been performed, and it is not known whether the values used in non-pregnant patients can be extrapolated to pregnant women. The absence of randomized clinical trials regarding thromboprophylaxis of pregnant women with COVID-19 leads to the heterogeneous aspect of therapeutic recommendations.

The clinical triad in pregnant women with COVID-19 is given by the damage to the respiratory, immunological (systemic inflammatory process), and hematological (CAC) systems. The most frequent hematological impairment of patients with COVID-19 hospitalized in the ICU was CAC, with distinct criteria from DIC and SIC. These differences are due to the high concentrations of D-dimers and fibrinogen, the prolonged coagulation time, and the lack of severe thrombocytopenia [59].

CAC is defined by the presence of thrombosis, bleeding, and changes in coagulation laboratory parameters [60]. The prevalence of CAC was 1% among all COVID-19 pregnancies, of which 65% required intensive therapy and 20% hospitalization, which indicates an evolution towards severe forms of this group of patients [37].

The CAC is characterized by fatal macro and microvascular thrombosis, multiple systemic organ failure, and death. The pathophysiological processes are represented by endotheliosis, immunothrombosis, exacerbated pro-inflammatory status, and changes in the coagulation system. The follow-up of patients with CAC is difficult, the monitored biomarkers are D-dimers and fibrinogen, and the standard therapy is heparin. The inflammatory marker angiopoietin-2 (Ang-2) is a possible biomarker that can evaluate the degree of endothelial damage. Medication with anti-interleukin-6 (tocilizumab) or anti-plasmin (nafamostat mesylate) has not proven its usefulness in these critical patients [61].

Thrombotic complications in patients with COVID-19 are caused by either pulmonary microvascular thrombosis (immunothrombosis) [62] or VTE [63]. Studies have not shown an increased incidence of VTE or immunothrombosis in patients with COVID-19. No VTE events were reported on treatment with prophylactic and therapeutic unfractionated heparin (UFH) or LMWH. Prothrombotic and fibrinolytic changes may last up to 3 months postpartum, then return to pre-pregnancy status [64,65].

Biomarkers used to predict the severity of the disease, and to a lesser extent CAC, are: increased levels of dimer D (a role for both coagulation and fibrinolysis); inflammatory markers of the acute phase (α1-acid glycoprotein, protein C-reactive, ferritin, procalcitonin, and high-sensitivity troponin); and decreased levels of factor V, VIII, and fibrinogen. Other possible markers are interleukin-6 and Ang-2, platelet count, APTT, PT, serum levels of lupus anticoagulant, antithrombin, and proteins C and S. These biomarkers may explain the pathophysiological mechanisms of CAC, having no real role in diagnosing or treating COVID-19 [66].

Jevtic et al. showed that 64% of COVID-19 infections were mild, while only 4% were severe, and 1% developed CAC. Changes encountered were increased C-reactive protein, D-dimer, thrombocytopenia, and lymphopenia. In 60% of cases with CAC, the standard dose of LMWH was used. However, the occurrence of VTE was observed in some patients regardless of prophylactic anticoagulation [37]. In CAC, the shortening of PT, the absence of changes in fibrinogen levels, mild thrombocytopenia, and the lack of schistocytes in the peripheral blood were observed, making the administration of fibrinogen or coagulation factors useless in this group of patients. The prognosis of CAC can be assessed based on elevated levels of fibrinogen, D-dimer, and von Willebrand factor, with near-normal levels of PT, aPTT, and platelet count [61].

Although there is a tendency to administer the standard prophylactic dose of LMWH (on admission or during hospitalization), the guidelines recommend the weight-adjusted prophylactic dose in the CAC patient group [39].

### 6.2. Disseminated Intravascular Coagulation

Frequently, the monitoring of a positive COVID-19 pregnant woman requires not only the assessment of the respiratory impairment but also of the potential complications with acute onset (DIC), this being achieved by investigating the coagulation tests and fibrinogen [59,67]. Servante et al. presented that from a series of 1063 pregnant women with COVID-19, regarding thromboembolic complications, 132 were admitted to intensive care, and 17 of them died, including two with DIC and two with PE [1].

SARS-CoV-2 viral infection with increased immunogenicity through the increase of Th17 cells causes a change in the ratio of Treg/Th17 cells, aggressively stimulating the systemic proinflammatory process, with the possible worsening of the patient’s condition [2]. Increases in D-dimers, PT, fibrinogen degradation products, hypofibrinogenemia, APPT, and thrombocytopenia occur in SARS-CoV-2 infection and DIC. The prothrombotic potential is increased in DIC, evidenced by the high titer of D-dimers. To predict the risk of DIC in pregnant women, an evaluation scale (ISTH DIC score) was created, whose score ≥26 points indicates DIC [68].

In 2022 Alhousseini et al. proposed the introduction of a score adjusted to the physiological and hematological changes during pregnancy, starting from the concept of non-overt DIC developed by ISTH in non-pregnant patients to early identify pregnant women at risk of developing DIC before the clinical onset, and to initiate as faster therapeutic management of these cases and reduce the mortality rate [69].

Special attention must be paid to pregnant women with asymptomatic/mild forms of COVID-19 in the 3rd trimester of pregnancy with risk factors regarding the possibility of developing a subclinical DIC (platelet count ≤100,000 per mm^3^, fibrinogen ≤200 mg/dL, and prothrombin time ≥3 s above the normal upper limit), in the conditions of decreased active fetal movements and a modified cardiotocographic trace. The guideline recommends evaluating coagulation tests and initiating transfusion protocols before delivery. Anticoagulant therapy is relatively contraindicated and dependent on assessing hemorrhagic risk [70].

The prognosis of pregnant women with COVID-19 is improved by using LWMH as antithrombotic therapy, except DIC with severe hemorrhagic diathesis, when the risks of administration outweigh the benefits. The administration of anticoagulant therapy must be safe for both the mother and the fetus [71].

## 7. Antiplatelet Therapy—Between Prophylaxis and Therapeutic Use

Antiplatelet therapy prevents clot formation and platelet adhesion and may improve the ventilation/infusion ratio in patients with COVID-19 and severe respiratory failure [72]. Low-dose prenatal aspirin administration is used in preeclampsia, thrombophilia, restriction of intrauterine growth of the fetus, and preterm birth. The International Federation of Gynecology and Obstetrics (FIGO) and the American College of Obstetricians and Gynecologists (ACOG) have stated that low-dose aspirin for prophylaxis against placental mediation pregnancy complications has no support [73,74].

Prophylactic administration of low-dose aspirin in pregnant women is indicated in those with moderate or increased risk of preeclampsia, with an incidence of 10–15% in all pregnancies [75]. Low-dose aspirin treatment reduces the risk of preeclampsia by 10% and the risk of fetal growth restriction by 20% and should be initiated between 12 weeks and 28 weeks of gestation and continued daily until delivery [73].

Aspirin at doses between 75 and 150 mg inhibits the production of isoenzyme cyclooxygenase 1, causing a decrease in platelet production of thromboxane A2, a potent vasoconstrictor. In the prophylaxis of preeclampsia, the benefit of aspirin use seems to outweigh the risk of adverse effects of COVID-19 infection [48].

Hereditary thrombophilias further increase the risk of maternal VTE [76]. Nahas et al. revealed that prophylactic treatment with enoxaparin in pregnant women (with or without thrombophilia) might increase the rate of live births [77]. Hamulyák et al., on 11 randomized controlled trials with 1672 pregnant women with persistent antiphospholipid antibodies and recurrent pregnancy losses, revealed a combination of heparin (unfractionated or LMWH) and aspirin during pregnancy may raise the live birth rate compared to aspirin alone [78]. Navaratnam enunciated the concept of “aspirin resistance” as a low response to the action of aspirin, either by decreased platelet activation or by the appearance of thrombosis during treatment [79]. In some cases, aspirin resistance requires either increasing the dose or the combination with LMWH at a dose of 40 mg once a day [80].

Until now, it has not been possible to distinguish between the chronic administration of antiplatelet therapy and the acute one with the confirmation of SARS-CoV-2 infection. During pregnancy, the chronic use of aspirin is primarily used in the prophylaxis of preeclampsia and in people with an increased risk of thrombosis. Meizlish et al., in a retrospective study of 2785 adult patients hospitalized with COVID-19, using multivariable regression analysis, established that a low mortality rate was obtained with anticoagulant therapy with either intermediate-dose LMWH or aspirin [81]. Although aspirin would be beneficial in the micro-thrombosis process, the study by Sahai et al. identified an increased risk of thromboembolic processes in patients with COVID-19 treated with chronic aspirin [82].

The utility of using antiplatelet therapy in patients with COVID-19 was contested in 4 randomized studies (RECOVERY, ACTIV-4a, ACTIV-4b, and REMAP-CAP), compared to observational studies, and it was recommended not to associate it with the standard anticoagulation scheme regardless of the severe forms of the disease. Randomized studies in pregnant women are also necessary because, in all these studies, pregnancy was an exclusion criterion [83].

## 8. Anticoagulant Treatment—The Border between Too Much or Too Little Efficacy

Following extensive research in the literature, we found very few references about the particularities of antithrombotic therapy in pregnant patients with COVID-19. The lack of clinical trials in pregnant women is explained by a series of reluctance on the part of patients to administer validating therapies and by the fear of specialists regarding therapeutic regimens whose pharmacokinetics are not fully known in pregnancy. Another aspect is related to the statistical significance of the study groups.

Many pregnant women are currently undergoing anticoagulant therapy for various indications. It is not fully known whether this therapy alone or in conjunction with other therapies plays a protective role. Regarding prophylactic doses, clinicians have reached a consensus on the safety of their administration in pregnancy, as opposed to using intermediate and therapeutic doses of anticoagulants, which must be performed by a multidisciplinary team that will analyze the timing and duration of anticoagulation. The prophylactic anticoagulant treatment for pregnant women with COVID-19 will depend on the gestational age, the time of delivery, the severity of the disease, and the underlying prothrombotic risk [63,84].

According to the European Society for Vascular Surgery 2021 guidelines on the Management of Venous Thrombosis, the first choice of anticoagulation in pregnancy should be LMWH, as it does not cross the placenta or breast milk [57]. Moreover, it is suggested that similar efficacy can be obtained with a single-dose regimen. In contrast, high doses of LWMH may increase the risk of bleeding without decreasing the risk of thrombosis. Unfractionated heparin and vitamin K antagonists are associated with a higher risk of hemorrhage throughout the pregnancy. Direct oral anticoagulants could increase teratogenicity and are formally contraindicated. While pregnant, women on oral anticoagulants should switch to LMWH. It can be useful to measure the blood levels of anti-Xa in pregnancy to maintain the desired therapeutic or prophylactic peak levels [85,86].

The ANTI-CO trial [86], PREVENT-HD study [87], ATTACC study [88], CHEST Guideline and Expert Panel Report [89], and DAWn-Antico study [90] represent a panel for antithrombotic therapy addressed for severe or critically ill patients with COVID-19. Extrapolating data regarding thromboprophylaxis from clinical trials in non-pregnant patients leads to major issues in establishing clear guidelines for treating pregnant women with COVID-19. According to the Pulmonary Embolism Response Teams, the most recommended therapy during the COVID-19 pandemic was anticoagulation, most patients being women with a history of DVT/PE [91].

A dilemma is related to modifying the antithrombotic treatment in pregnant women who follow anticoagulant therapy and are suspected of COVID-19. In pregnant women with COVID-19, the risk of VTE increases with the severity of the disease. In all symptomatic patients with COVID-19, thromboprophylaxis should be performed until complete recovery if there are no contraindications to heparin. The anti-inflammatory effect of heparin is based on the attachment to proinflammatory cytokines (interleukin-6), inhibiting neutrophil chemotaxis and leukocyte migration. Tang et al. suggested that some patients may benefit from heparin prophylaxis. They reported that patients with a sepsis index ≥4 or an increased level of D-dimer >3 µg/L had lower mortality with heparin prophylaxis [92].

Retrospective studies on oral or injectable anticoagulant treatment before admission can provide important information on how treatment and complications are due to COVID-19. Denas et al. have shown that chronic treatment with oral anticoagulants in pregnant women for different diseases decreases the risk of cardiovascular thromboembolic events and thus decreases maternal morbidity and mortality rates [93]. Another study has shown that anticoagulant treatment in the acute phase may benefit selected patients [92]. However, chronic anticoagulant treatment was not found to decrease the risk of morbidity and mortality associated with SARS-CoV-2 infection [31,94]. The role of chronic anticoagulation should be confirmed in subsequent studies.

A possible side effect of prophylactic therapy with heparin would be thrombocytopenia, which may be associated with increased mortality from COVID-19. In a meta-analysis (8 studies and 2946 patients), Abdel-Maboud et al. described that prophylactic heparin did not significantly influence the mortality rates in patients with moderate COVID-19. Prophylactic heparin administration in patients with mild symptoms may positively affect the outcome if combined D-dimers > 3 µg/L, platelet count > 100 × 109/L, and PT < 14 s, regardless of comorbidity, sex, or age [95]. Another study revealed the potential role of heparin in reducing the mortality rate [92]. In a randomized, open-label, phase II study (HESACOVID), Lemos et al. observed a statistically significant increase in the partial pressure of arterial oxygen (PaO_2_)/fraction of inspired oxygen (FiO_2_) ratio in the enoxaparin group versus prophylactic group [96].

There are some patients with COVID-19 with a higher incidence of thrombotic events associated with bleeding episodes, so it is difficult to perform prophylactic anticoagulant therapy. Despite thromboembolic prophylaxis with high doses of LMWH, Chistolini et al. observed a reduced incidence of thrombotic complications without increasing the risk of bleeding [44]. Prophylactic administration of LMWH in microvascular thrombosis has been recommended by the World Health Organization, the ISTH, and the American Society of Hematology without setting the standard dose. In conclusion, the international recommendations advocate for prophylactic doses and not for intermediate and therapeutic doses of LMWH [73,97,98].

According to the ISTH recommendations on the treatment of coagulopathy in COVID-19, initiation of prophylactic anticoagulant therapy with LMWH should be performed as soon as possible to prevent extensive thrombosis [33]. The balance between the risk of thrombosis and the possible risk of bleeding is essential to evaluate the beneficial non-anticoagulant effects of high-dose LMWH [99,100,101] (Figure 2).

In conclusion, the current evidence is insufficient to support the role of prophylactic heparin in reducing mortality in patients with COVID-19. There is a need for other randomized controlled studies, with stratification of patients according to D-dimers, PT, and platelet count levels.

## 9. Thromboprophylaxis in Pregnant Women with COVID-19

Hemostatic changes in COVID-19 coagulopathy are either the consequence of a severe inflammatory process or the specific virus-mediated effect. The immune response to acute SARS-CoV-2 infection, followed by increased cytokines and inflammatory mediators, triggers thrombogenesis. The initial strategy to reduce inflammation can prevent thrombosis [98].

Although there are guideline recommendations to address the management and prognosis of coagulopathies related to COVID-19 during pregnancy, the therapeutic strategy regarding thromboprophylaxis in pregnancy is not fully resolved [102]. Currently, the mechanisms of action of SARS-CoV-2 infection at the cellular level are incompletely known, stating a possible interaction between thrombotic, immunological, inflammatory, and coagulation factors. Thrombotic complications can be initial, concomitant, or consecutive to immunological ones. Thus, strategies for thrombo-prophylaxis or immuno-prophylaxis are currently possible ways to prevent and treat the worsening of the clinical picture in patients with COVID-19 [102].

D-dimers normally increase progressively during pregnancy and peak in the third trimester. A marked increase in D-dimers above the normal pregnancy level (median in third trimester 1.24 μg/mL) [103] indicates possible COVID-19 coagulopathy or DIC [39]. Usually, thromboprophylaxis in pregnant women with COVID-19 is performed with LMWH. The initiation and duration of prophylactic anticoagulation in pregnancy and COVID-19 should consider the severity of the disease, the time of birth relative to the onset of the disease, the underlying prothrombotic risk secondary to comorbidities, and the presence of major bleeding from obstetric or non-obstetric causes, including coagulopathy [104]. The elective treatment used in VTE does not act on immuno-thrombosis. The occurrence of thrombotic phenomena in patients undergoing prophylactic or curative therapy with LMWH should consider anti-cytokine and antiviral therapy for the process of immunothrombosis [3].

Antiplatelet therapy with low-dose postpartum aspirin is not yet fully elucidated regarding its real role in preventing thromboembolic complications. A prophylactic standard dose of LMWH has been recommended by the ISTH. Despite using a standard dose of LMWH in patients with COVID-19, Miesbach et al. revealed a higher incidence of VTE in more than 40% of cases [97]. The currently low incidence of severe cases of COVID-19 in pregnant women can be explained by the disproportionate increase in those who use LMWH prophylaxis for various conditions compared to the accurate indications, the recommendations of vaccination of pregnant women, a higher rate of hospitalization, and the correct organization of health services [4].

Middleton et al., in a meta-analysis of 29 studies (involving 3839 women), showed that data on the benefits and risks of thromboprophylaxis in pregnant women at high risk of VTE during pregnancy and postpartum are disparate and inconclusive due to the lack of clinical trials and the existence of only a series of cases with a small number of participants [105]. Khalil et al., in a meta-analysis of 17 studies involving 2567 pregnancies, observed an increase in the rate of admission to the intensive care units in patients with comorbidities and those over 35 years old. An iatrogenic increase in preterm birth and cesarean delivery was also observed, with a maternal mortality of ~1% and perinatal mortality below 1% [106].

Ena and Valls observed that therapeutic or prophylactic anticoagulation with LMWH did not alter the death rate or the institution of invasive mechanical ventilation. The therapeutic dose in the non-critical patient with COVID-19 is indicated in patients with high thromboembolic risk and low hemorrhagic risk [107]. Servante et al. identified a higher rate of hematological complications in pregnant women with COVID-19 than those without infection (1.26% versus 0.45%). According to the RCOG, all pregnant women hospitalized or not with confirmed or suspected SARS-CoV-2 infection benefit from the prophylactic dose of LMWH, except for giving birth less than 12 h after hospitalization, and this continues for 10 days after discharge. If the pregnant woman with COVID-19 is close to giving birth, an important role regarding the clinical implications is represented by the coagulation analyses and the number of platelets [1]. 

One hypothesis would be that pregnant women with COVID-19 who have received chronic anticoagulant treatment for various conditions appear to have a lower risk of morbidity than non-pregnant patients. This hypothesis will have to be confirmed in further studies.

## 10. Discussion

Due to the increased risk of severe respiratory damage and other complications, pregnant women are a population at higher risk of COVID-19 compared to the general population. Pregnant women with comorbidities or chronic diseases have a significantly increased risk of developing complications [108]. As a result, the strategy of healthcare systems must focus on the pregnant woman and fetus, with the development of clear guidelines on the prevention and management of COVID-19 thrombosis and coagulopathy. This is difficult, as there are few studies on pregnant women.

The SARS-CoV-2 infection during pregnancy can affect both the mother and fetus. The rate of complications depends on gestational age and is associated with the risk of preterm birth, intrauterine growth restriction, neonatal intensive care unit admission, and perinatal complications [13]. Knowledge of COVID-19 involves the epidemiological, thrombotic, and immunological interpretation of the mode of action, the mechanisms underlying coagulopathy, and treatment regimens. Studies have shown an increased risk of thromboembolic disease in pregnant women with or without pharmacological thromboprophylaxis in standard doses [5,107].

The etiopathogenic mechanisms of coagulopathy caused by acute SARS-CoV-2 infection consider the specific effect mediated by the virus and the severe inflammation accompanied by hemostatic disorders. The degree of involvement of the two mechanisms is not fully elucidated, with the increased risk of thrombosis resulting from the action of cytokines and inflammatory mediators [8,40]. A possible therapeutic strategy would be related to thrombosis prophylaxis by reducing inflammation. Another hypothesis regarding the pathophysiology of coagulation was related to the kallikrein-bradykinin pathway. The addition of a recombinant interleukin-1 receptor antagonist (anakinra) may play an essential role in the excessive thromboinflammatory response to COVID-19 [90].

Limited and usually retrospective clinical studies mean that the prevalence of COVID-19 thrombosis is not known, even less in pregnant women, requiring more extensive studies in the future.

Severe forms of COVID-19 in pregnancy worsen the fetal prognosis by increasing the prothrombotic effect at the placental level by identifying areas of infarction, vascular malperfusion, foci of thrombosis, and infiltrates secondary to acute and chronic inflammation, which causes a major risk of stillbirth. In such cases, the cesarean section rate increases, and the perinatal results show a worse prognosis for these fetuses [109]. Alteration of maternal–fetal vascular perfusion and thrombosis of greater vessels of the fetus have been observed by analyzing placentas in patients with COVID-19. This situation occurs due to the hypercoagulable or increased inflammatory state, without inflammatory or infectious changes being evident at the placental level [110]. 

Since most severe forms of SARS-CoV-2 infection in pregnant women requiring hospitalization were observed in the third trimester of pregnancy, researchers showed that vaccination before the 30th week of pregnancy is necessary [111]. The different clinical evolution of COVID-19 in pregnant women is due to different viral variants (currently the dominant variant being Omicron), viral aggression, the degree of vaccination of the female population of reproductive age, recurrent forms, and comorbidities [109,112].

The main risk factors regarding maternal morbidity are antepartum admissions related to COVID-19, maternal age, obesity, and comorbidities. The low incidence of complications and precocious-initiated thromboprophylaxis in pregnant women with COVID-19 were related to a higher admission rate than in non-pregnant patients [36].

Currently, there is insufficient data on using intermediate or therapeutic doses of LMWH. In pregnant women with mild to moderate forms of COVID-19, the use of prophylactic doses is recommended, not to significantly reduce the risk of thrombosis, but to maintain an increased risk of bleeding. Severe forms continue to benefit from using therapeutic doses of LMWH [5,51,52,53,54,55,56,57,58] (Figure 3).

There are no clear data on whether prophylactic anticoagulant therapy could provide a protective benefit in COVID-19. There is insufficient and contradictory evidence on using low-dose aspirin in thromboprophylaxis [81,82]. Some studies have found a lower mortality rate in patients treated with low doses of aspirin [81,113].

## 11. Limitations

A limitation is given by the lack of randomized clinical trials to the relatively small number of patients included in research studies for developing clinical guidelines. The duration of the pandemic and the possible appearance of some strains that generate changes in clinical behavior can also be limitations of this study.

## 12. Conclusions

Pregnant women with COVID-19 are a special category of patients with an increased risk of VTE. There is currently clinical variability in measuring biomarkers and administration of anticoagulant therapy, in the outpatient setting, during hospitalization, and after discharge. An individualized decision must carefully establish the optimal prophylactic and therapeutic anticoagulant regimen in each patient. Future research is needed to select the appropriate anticoagulant dose and duration of anticoagulation in pregnant women with COVID-19.

## Figures and Tables

**Figure 1 ijerph-20-01949-f001:**
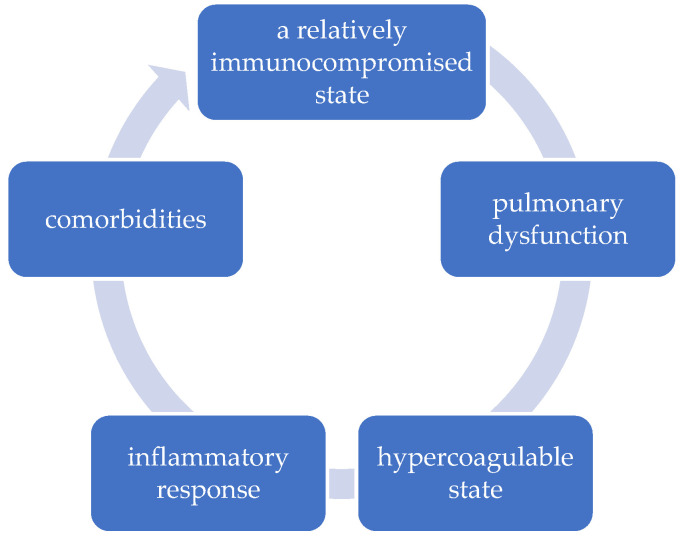
The vicious circle of etiopathogenic factors in pregnant women with COVID-19.

**Figure 2 ijerph-20-01949-f002:**
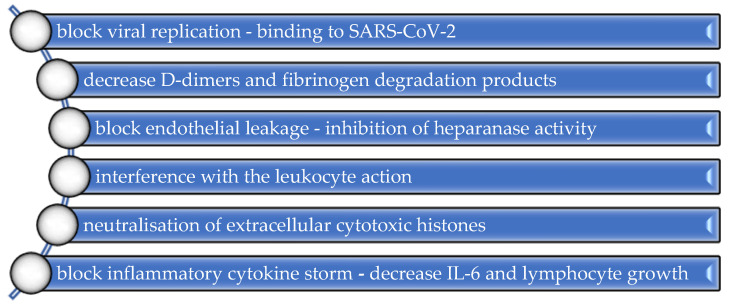
Non-anticoagulant properties of LMWH.

**Figure 3 ijerph-20-01949-f003:**
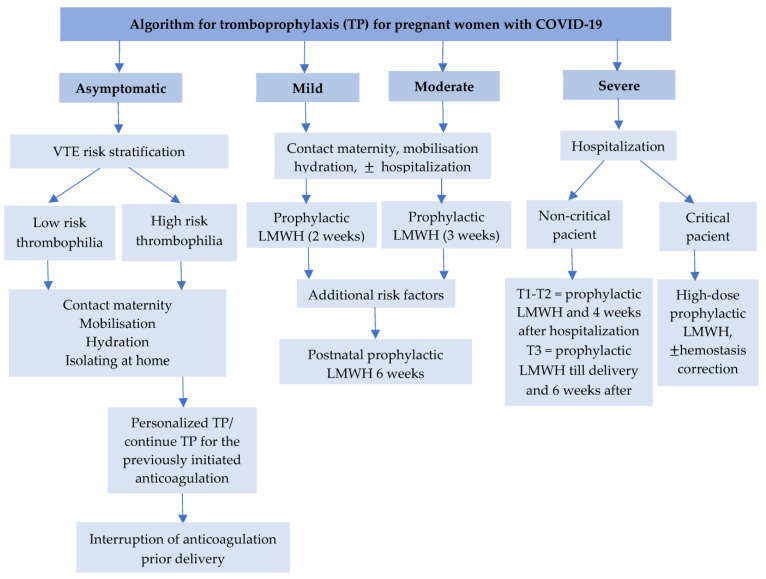
Algorithm for thromboprophylaxis for pregnant women with COVID-19. (T1, T2, T3—first, second, and third trimester, VTE—venous thrombembolism, LMWH—low molecular weight heparin).

**Table 1 ijerph-20-01949-t001:** Synopsis of guidelines from international societies on thromboprophylaxis in pregnant women with COVID-19.

*Guidelines*	
*RCOG* [51]	Antenatal assessment - Self-isolating at home Hydration and mobilization If the VTE risk score is >3, prophylactic LMWH continued until recovery 7–14 days - Hospitalized VTE prophylaxis should be prescribed during admission unless contraindicated or birth expected within 24 h On high-flow oxygen, CPAP, non-invasive/invasive ventilation—dose LMWH Postnatal assessment - Conduct VTE risk-assessment following birth - For those with confirmed SARS-CoV-2 infection, prescribe prophylactic LMWH for 10 days
*ACOG* [52]	Antepartum period - VTE diagnosed during pregnancy—adjusted-dose LMWH/UFH - History of single unprovoked VTE—prophylactic, intermediate-dose, or adjusted-dose LMWH/UFH - Low-risk thrombophilia Without previous VTE—surveillance without anticoagulation therapy With a family history of VTE—surveillance without anticoagulation therapy or prophylactics LMWH/UFH with a single previous episode of VTE—prophylactic or intermediate-dose LMWH/UFH - High-risk thrombophilia Without previous VTE—prophylactic or intermediate-dose LMWH/UFH With a single previous episode of VTE or an affected first-degree relative—prophylactic, intermediate-dose, or adjusted-dose LMWH/UFH Postpartum period - VTE diagnosed during pregnancy—adjusted-dose LMWH/UFH for a minimum of 6 weeks - History of single unprovoked VTE—prophylactic, intermediate-, or adjusted-dose LMWH/UFH regimen for 6 weeks - Low-risk thrombophilia Without previous VTE—surveillance without anticoagulation therapy or prophylactic anticoagulation if the patient has additional risk factors with a family history of VTE—prophylactic anticoagulation therapy or intermediate-dose LMWH/UFH With a single previous episode of VTE—prophylactic anticoagulation therapy or intermediate-dose LMWH/UFH - High-risk thrombophilia Without previous VTE—prophylactic anticoagulation therapy or intermediate-dose LMWH/UFH With a single previous episode of VTE or an affected first-degree relative—prophylactic anticoagulation therapy or intermediate or adjusted-dose LMWH/UFH for 6 weeks
*CNGOF* [53]	Risk will be analyzed according to personal risk factors and oxygen requirement: - Low risk = lack of prophylaxis - Medium risk = standard LMWH prophylaxis - High risk = higher prophylactic doses with LMWH The duration of prophylaxis must be maintained until recovery and will not start if delivery is approaching.
*COVID Collaborative Group, Barcelona* [54]	Antepartum period - Self-isolating at home In patients with infection >4 weeks before delivery—standard thromboprophylaxis Avoid immobilization - Hospitalized Prophylactic LMWH and 2 weeks after that Postpartum period Prophylactic LMWH during hospitalization and 6 weeks after in cases with severe COVID-19
*NIH* [55]	Antepartum period - Continuation of anticoagulant or antiplatelet therapy for underlying conditions in the absence of contraindications - Prophylactic doses of anticoagulation for pregnant women who are hospitalized with confirmed COVID-19 - Post-discharge VTE prophylaxis is not routinely recommended in pregnant patients; individualization is according to concomitant VTE risk factors. Postpartum period - Anticoagulant therapy in patients who require VTE prophylaxis or treatment
*ASH* [56]	Antepartum period - Suggests against intermediate-dose LMWH prophylaxis compared with standard-dose LMWH prophylaxis Postpartum period - Suggests either standard- or intermediate-dose LMWH prophylaxis
*ESVS* [57]	In DVT, therapeutic doses of LMWH are recommended during pregnancy for at least 12 weeks and postpartum for at least 6 weeks.In DVT less than 2 weeks before the due date, a temporary filter can be used in the inferior vena cava
*ISTH* [58]	Anticoagulant therapy with LMWH was accompanied by a better prognosis in forms that meet SIC criteria or markedly increased D-dimer.

RCOG—Royal College of Obstetricians and Gynaecologists, ACOG—American College of Obstetricians and Gynaecologists, CNGOF—French National College of Obstetricians and Gynecologists, NIH—National Institutes of Health, ASH—American Society of Hematology, ESVS—European Society of Vascular Surgery, ISTH—International Society on Thrombosis and Haemostasis, VTE—Venous ThromboEmbolism, LMWH—Low Molecular Weight Heparin, DVT—Deep Venous Thrombosis, SIC—Sepsis-Induced Coagulopathy, UFH—Unfractionated Heparin, CPAP—Continuous Positive Airway Pressure.

## Data Availability

Not applicable.

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
