# Peer review of "Thromboprophylaxis in Pregnant Women with COVID-19: An Unsolved Issue"

_ijerph, 2023, doi:10.3390/ijerph20031949_

Round 1

Reviewer 1 Report

Well done, only two minor comments.

1) page 3, line 96. You describe pregnancy outcomes for COVID patients regarding maternal deaths and early neonatal deaths. Did you find anything regarding stillbirths?

2) page 12, line 465. Looks like there is a reference missing.

Author Response

Dear Reviewer,

I revised it according to your recommendations. 

  • page 3, line 96. You describe pregnancy outcomes for COVID patients regarding maternal deaths and early neonatal deaths. Did you find anything regarding stillbirths?

Answer: Thank you for your mention; we added data regarding the incidence of stillbirth. (please see the attached manuscript - line 111).

  • page 12, line 465. Looks like there is a reference missing.

Answer: Thank you for your remark; we fixed it. (please see the attached manuscript - line 476).

Kindest regards

Reviewer 2 Report

The article is consistent within itself. The references are relevant and recent. The cited sources are referenced correctly. Appropriate and key studies are included. The paper is comprehensive, the flow is logical and the data is presented critically.

However, there are some specific comments on weaknesses of the article and what could be improved:

Specific comments on weaknesses of the article and what could be improved:

Major points - none

Minor points

1. The search strategy for papers to be cited could be included in the paper after the introduction.

2. Please, state the limitations of this narrative review.

Author Response

Dear Reviewer,

I revised it according to your recommendations. 

  1. The search strategy for papers to be cited could be included in the paper after the introduction.

Answer: Thank you for your mention; we added data regarding the incidence of stillbirth. (please see the attached manuscript - lines 77-89).

  1. Please, state the limitations of this narrative review.

Answer: Thank you for your recommendation; we added limitations.

(please see the attached manuscript - lines 607-611).

Kindest regards
